# Determinants of Digital Transformation in the Hospitality Industry: Technological, Organizational, and Environmental Drivers

Maria Nikopoulou [1] , Panos Kourouthanassis [1], Giota Chasapi [1], Adamantia Pateli [1] and Naoum Mylonas [2,*]

1   Department of Informatics, Ionian University, Tsirigoti Square 7, 49100 Corfu, Greece
2   Department of Tourism, Ionian University, P. Vraila Armeni 4, 49132 Corfu, Greece
*   Correspondence: nmylonas@ionio.gr

**Abstract:** The current study aims to investigate the factors that affect a hotels' decision to adopt digital technologies. Our theoretical grounding builds on the Technology–Organization–Environment (TOE) research framework. Our research model was validated through a survey of 502 hoteliers and managers using the Partial Least Squares–Structural Equation Modeling (PLS-SEM) statistical method. The results indicated that micro, small and medium-sized enterprise (MSME) hotels affected by the COVID-19 outbreak are more likely to adopt digital technologies. The intention to adopt digital technology is positively and significantly influenced by the digital maturity of organizations, financial resource availability and government regulations. The current study investigates rather less explored factors, such as the organizational digital maturity, which consists of a multi-dimensional latent variable. Our findings may be employed to guide the formulation of digital strategies by hospitality industry organizations.

**Keywords:** digital technologies; TOE framework; hospitality; digital maturity; COVID-19; digital transformation

## 1. Introduction

The COVID-19 pandemic has had a long-lasting impact on a global scale, and the hospitality industry has become one of the most financially jeopardized industries [1]. Because of the COVID-19 outbreak, hotels have revisited their digital strategy. For example, to avoid direct contact with their guests, hotels have adopted touchless mobile applications, such as mobile keys, in order to bypass interaction with the reception desk [2].

The hospitality industry had historically been reluctant to adopt and implement digital technologies [3]; micro, small and medium enterprise (MSME) hotels have been reported as late adopters of digital technology solutions [4]. Nevertheless, scholarly attention towards understanding facilitators and inhibitors of digital technology adoption has increased, especially as a means to tackle the challenges imposed by the pandemic [5]. In addition, the latest information system (IS) literature on accessing an organization's state of digitalization is investigating its digital maturity [6]. Digital maturity is defined 'as the status of a business digital transformation' [7]. Digital transformation is possible by the adoption of digital technologies [8]. Interestingly, there is only a limited number of studies investigating the technology adoption of MSME hotels [9,10].

Based on the above, this study developed a comprehensive framework to investigate the technological, organizational, and environmental factors that affect hoteliers' and managers' decisions to adopt digital technologies. More specifically, the study is aiming to explore the following research questions:

(1)   What are the technological factors influencing hoteliers and hotel managers to adopt digital technologies?

(2) What are the organizational factors influencing hoteliers and hotel managers to adopt digital technologies?

(3) What are the environmental factors influencing hoteliers and hotel managers to adopt digital technologies?

To answer the research questions, the technological–organizational–environmental (TOE) theoretical framework is adopted [11,12]. Hence, by using the TOE framework and applying partial least squares (PLS)– structural equation modelling (SEM), the hypotheses on a dataset of 502 hotels in Greece were tested.

The research presents significant revelations in the technology and management literature. There has been general interest among scholars in digital technology adoption such as robotics and AI conducted from the guest's perspective [13,14]. On the other hand, these technologies from the managers' standpoint remain under-researched [13]. Thus, added knowledge is provided to hotel owners and managers. Furthermore, this study investigated a less explored but rather important technology adoption antecedent, namely organizational digital maturity under the context of hotels' culture, organizational structure, attitude towards technology, and organizational insights.

The current study is organized into seven sections. Section 2 reviews the literature on digital technology adoption in the hospitality industry as well as background information concerning digital maturity and the TOE framework. Section 3 develops the research framework and hypothesis development. Section 4 describes the research methodology. Section 5 shows the results of the empirical study, including the measurement and structural model. Section 6 presents the discussion of the research outcomes. Section 7 describes the theoretical and practical implications of the findings. Section 8 concludes with the limitations of the research and the avenues for future research. Lastly, in Section 9, conclusions are presented, which summarize the study and list the main obtained results.

## 2. Theoretical Background

### 2.1. Digital Technologies Adoption in the Hospitality Sector

Traditionally, the hospitality industry had been characterized as resistant towards technology adoption [3]. Nevertheless, many hotels have recently recognized their contribution to their overall performance and long-term prosperity. The COVID-19 pandemic had forced technology adoption, and this has resulted in the transformation of the tourism and hospitality industry from 'high-touch' and 'low-tech' to 'low-touch' and 'high-tech' [15]. The digital transformation journey of MSMEs in the hospitality sector started from the reorganization of websites, which were employed as the primary channel with their clients during the pandemic by sharing information about their COVID-19 policy and hygiene standards [16]. Prior studies have revealed that websites create the first all-important impression of the hotel and could increase the direct bookings and build a hotel's brand [17].

Furthermore, mobile and remote-control applications have witnessed a wider use [18]. For example, some studies have reported that a vast majority of clients in China pay their hospitality bills via their mobile phones [19]. The pandemic increased the adoption and use of mobile technologies by hotels' guests. For example, Hilton employed a contactless check-in that uses face recognition software. Social media platforms represent another category of digital technology that is widely used in the hospitality sector to achieve direct communication with customers [20] and as a strategic tool for achieving a reduction in operational costs, improving clients' relationship and loyalty [21].

In terms of interorganizational systems, customer relationship management (CRM) systems have enabled hotels to gather and bring forth customer knowledge across different and multiple points of contact in order to obtain comprehensive knowledge about consumer behavior and needs [22]. During the pandemic, CRM was shifted to a vitally important tool for hotels to build long-term relationships with guests, which further impacts their innovation capability [23]. The hospitality sector also closely monitors technology developments, including artificial intelligence (AI) and robotic technology [24]. For example, the Henn-na Hotel in Japan introduced the first fully robotic-service-automated hotel world-

wide [25] with studies reporting an increase in the acceptance of service robots, shifting service interactions towards purely technological communications [26]. Recently, hotels have started to investigate the adoption of chatbots as a means of automating customer interactions and responding to their requests [27].

A great number of studies have highlighted the significance and implications of digital technologies in the hotel sector. Additionally, many scholars argue that the pandemic is a technological opportunity not only to overcome the severe health crisis but also to prepare for future challenges [28].

### 2.2. Capturing the Digital Maturity of Organizations

Digital maturity is defined as the ability of an organization to systematically and efficiently adopt the ongoing digital changes through management practices, and it also refers to the digital transformation journey. [6,29]. Digital maturity encompasses two interweaved dimensions. From a technological stance, digital maturity reflects the extent to which organizational tasks are supported by information technologies. From a managerial stance, digital maturity reflects the organizational transformation efforts (i.e., upskilling or reskilling of employees, attitude of high management towards information technologies, redesigning business processes, and so on). Prior examination of hotel technology research revealed that scholars have given great attention to investigating the digital maturity of guests [30] and/or hotel employees [31].

Digital maturity is an emerging concept; hence, scholars have just started to propose pertinent measurement models. Thordsen et al. [6] review extant digital maturity models. Their findings indicate that digital maturity organizations should be perceived as a multi-dimensional concept that involves cultural, organizational, technical, and managerial aspects of the firm. For example, technological elements may encompass hardware and software investments, whilst organizational elements may address human resources information technology skills and know-how [32]. Likewise, organizational factors may include the culture of the firm, as well as the firm's strategy towards the human capital [33]. A similar perspective is reported by Kane et al. [34], who relate the digital maturity of forms with aspects related to organizational digital capabilities, strategies, culture and talent and skills. Under this prism, digital maturity is more of an organizational challenge rather than a technical one since technology developments usually outpace the ways organizations evolve over time [35].

Scholars may aggregate and estimate the current digital maturity level of the organization under investigation by capturing the individual performance of the firm in each dimension [36]. Interestingly, not all organizations follow the same trajectory of digital maturity throughout their digital transformation journey. Instead, digital maturity is context-specific and may follow idiosyncratic avenues [37].

Among the plethora of tools to measure digital maturity, the current study considers that the recognition of the digital maturity variables beyond the individual characteristics could capture a deeper understanding of the digital technology adoption process as multi-dimensional factors that comprise the technological construct are investigated.

### 2.3. The Technology–Organization–Environment (TOE) Framework

The TOE framework portrays the idea that digital technologies shall be investigated by spotlighting the organization as an entity [11,12]. More analytically, the TOE framework is an organization-level theory that explains three different elements of a firm's context influence adoption decisions, namely the technological, organizational and environmental context [12]. Exactly this inclusion of these contexts manages to set TOE as the most suitable theory for scrutinizing technological adaptability, technology usage and generation of value from technological novelties [38]. Concerning the technological context, it outlines the internal and external technologies relevant to a firm [11]. In regard to the organizational context, it refers to depictive measures connected to firms, referring to the firm's scope, size, financial resources and administrative beliefs [39]. They are characterized as pivotal

to a firm's adoption of technological novelty. Lastly, the environmental aspect examines the environment in which a company is active [40]. It focuses on the environment in which a firm carries out its business transactions, prioritizing external aspects swaying the industry, such as government incentives and regulations [41].

It should be noted that the technological dimension of the TOE framework has been widely studied by scholars using Rogers' Diffusion Theory [42]. Previous literature investigation revealed that the most influential factors concerning the technological context of the TOE framework to be relative advantageous are compatibility and complexity [14]. Those are characterized as the ultimate factors that influence the digital technology adoption within the organizations [12]. In effect, scholars suggest that cultural values are important in technology readiness [43] and determine that the cultural values of the firm as the cultural backgrounds of the hotel employee's perspective [31]. Chi [23] highlighted that to optimize digital technology adoption, knowledge-sharing strategies should be developed from managers in their decision-making process.

## 3. Hypotheses Development and Conceptual Model

The research aims to develop a model that identifies six determinants of digital technology adoption in the hospitality industry within the three dimensions of the TOE framework: digital maturity, including 'culture', which deals with empowerment of employees with digital technology, 'organization', which is the use and adoption of technology, 'technology', which is the digital strategy, and 'insights', which involves the customer and business data to measure strategy success; the organizational context, which involves 'financial resources availability' to fund business initiatives; and the environment context, which is 'government regulation'. The current study investigates rather less explored factors and posits that the hotels are more likely to adopt digital technologies when their digital maturity is achieved.

### 3.1. Technological Context

The digital maturity of a firm was found to motivate and propel towards technology adoption [44]. Prior research has demonstrated that digital maturity can be linked to customer satisfaction [45] and perceived service quality [46]. According to Parasuraman [47], technology readiness, a closely linked term with digital maturity, may strongly affect the probability that people will embrace new technological systems and services. Technology readiness has been linked to technological acceptance [48]. The COVID-19 pandemic has made management more aware of technological solutions. As Effendi et al. [49] stated, during the pandemic, the awareness and acceptance of social media was a determinant of technology adoption. On this basis, the following hypothesis is proposed:

**Hypothesis 1. (H1):** *The digital maturity of organizations has a positive impact on digital technology adoption.*

It has already been discussed that technological acceptance can have a positive effect on technology solutions. According to Lam et al. [48], technology acceptance itself is linked to parameters of digital maturity connected to the culture of a firm. A variety of studies have linked digital maturity positively with the empowerment of employees in the usage of technology solutions [21]. Moreover, employees that work in a supportive climate that influences and provides strategic directions for the adoption of digital technology are more likely to adopt it [50]. According to the above, the following hypothesis is proposed:

**Hypothesis 1a. (H1a):** *The 'culture' element of digital maturity has a positive effect on digital technology adoption.*

According to Li et al. [46], hospitality industries require a workforce that is able to utilize modern technological tools to better serve their customers. Since new technologies

are often introduced, effective hospitality management needs to find ways to provide training to its employees in order to constantly provide high-quality technology solutions and gain the trust of its customers [30]. Moreover, the level of digital technology adoption is refined from the sophistication level of the firms' digital strategy [49]. According to the above, the following hypothesis is proposed:

**Hypothesis 1b. (H1b):** *The 'organization' element of digital maturity has a positive effect on digital technology adoption.*

The 'technology' element involves the company's adoption and usage of modern technology architectures within its business operation [44]. In this concept, the technology budget is important to allow the digital adoption [36]. Thus, the managers' budgetary participation is linked with their willingness for digital adoption within the hotel [51]. In addition, Sunny [31] suggested that the perceived long-term benefits of the adoption and usage of the digital technologies could be considered as a positive impact on digital transformation within the hotel sector. The 'technology' dimension also takes into account the customers' experience and feedback as an asset in order to develop the future technology design [36]. Hence, the identification of the customer's perception towards using digital technologies is a pivotal element for its adoption. The potential advantages of the adoption and implementation of technology advances could not be achieved if the hotel guests do not favor and appreciate these technologies [52]. According to the above, the following hypothesis is proposed:

**Hypothesis 1c. (H1c):** *The 'technology' element of digital maturity has a positive effect on digital technology adoption.*

The 'insights' latent variable of the technology context includes clear and quantifiable goals for measuring the success of the firms' digital strategy [44]. Therefore, setting clear goals for measuring the success of digital technology adoption will enable them to operate their business effectively or encourage future development [53]. Thus, it is critical that every employee understands how its performance is tied to corporate digital goals [44]. Overall, the existing literature highlights the pivotal role of technology-savvy leaders in the digital technology adoption and implementation procedure [54]. Moreover, in the 'insights' variable, the customer feedback is a critical variable in the digital technology adoption process [36]. Accordingly, the success of digital technology adoption depends on the organization's strategic use of the customer, business data and technology [36,55]. Consequently, the below-mentioned hypothesis is proposed:

**Hypothesis 1d. (H1d):** *The 'insights' element of digital maturity has a positive effect on digital technology adoption.*

*3.2. Organizational Context*

Organizational readiness, as used in previous studies on technology innovation adoption, assesses if the firm has sufficient financial resources [56,57]. In addition, Iacovou et al. [57] suggested that financial resource availability refers to available funds for the adoption and implementation of digital technology. Thus, firms that devote greater financial resources to digital technology adoption are more likely to achieve digital technology adoption. Concerning the COVID-19 crisis, despite the government response to guarantee businesses' existence through economic stimulus packages, hoteliers reconsidered financial adaptability [58]. Hence, according to the above, the following hypothesis is proposed:

**Hypothesis 2. (H2):** *'Financial resource availability' has a positive impact on digital technology adoption.*

### 3.3. Environmental Context

Environmental context refers to the area in which a business conducts its operation, including government regulations [11,12]. Government regulations refer to government encouragement or restrictions [59]. At the time of the COVID-19 pandemic, along with regulatory support, government restriction was also applied to minimize the impact towards casualties [60]. Zhu et al. [61] suggested that the effect of government pressure is critical within TOE framework. According to Salwani et al. [41], government regulations are critical in the adoption and implementation of digital technology, and as Baker [62] revealed, when governments impose constraints in the industry, digital technology adoption can be forced. Therefore, the following hypothesis is proposed:

**Hypothesis 3. (H3):** *'Government regulations' have a positive impact on digital technology adoption.*

The proposed hypothesized conceptual research model is illustrated in Figure 1.

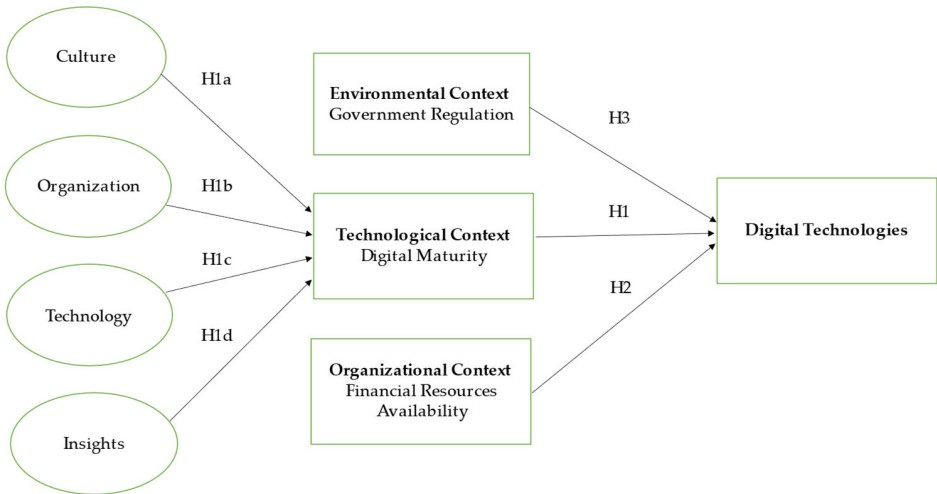

**Figure 1.** Hypothesized conceptual framework.

## 4. Research Method

### 4.1. Data Sample

The current research was conducted under the auspices of the Greek Ministry of Tourism. All members of the Hellenic Chamber of Hotels were invited to join the study. The criteria for selecting the sample were limited to managers and hoteliers since they are considered the ultimate decision-makers in the businesses [14]. The final sample consists of 502 hotels in Greece out of 5800 in total. The response rate was around 10%. For the current study, the research instrument adopted was an online questionnaire consisting of single-select multiple-choice, close-ended answers with one dichotomous question. Therefore, as the COVID-19 pandemic forced strict social distancing, the only considerable, faster and available way to contact the large volume of respondents across Greece was via an online questionnaire. To avoid non-response to any questions, the online questionnaire allowed no further processing when a question was not answered. An introductory note was included in the online questionnaire in order to introduce the purpose and objectives of the study to the respondents. The respondents were also informed that their identities were confidential and their participation was voluntary. All the data were securely saved. The questionnaire was completed by hotel owners and managers. The respondents completed a "five-point Likert scale" ("1 = strongly disagree" to "5 = strongly agree") [63]. All items are provided in Appendix A. With regard to the demographics, the vast majority (97.4%) were micro (48.8%), small (34.1%) or medium-sized enterprises (14.5%). A summary of demographics is presented in Table 1.

**Table 1.** Sample Demographics.

| Demographics | Frequency | Percentage (%) |
|:---:|:---:|:---:|
| **Position** | | |
| Owner | 278 | 55.4 |
| CEO | 134 | 26.7 |
| Senior executive | 58 | 11.5 |
| Other | 32 | 6.4 |
| **Hotel Category** | | |
| Independent Hotel | 430 | 85.6 |
| Member of big chain hotel | 20 | 4.0 |
| Member of a small chain hotel | 52 | 10.4 |
| **Geographical region** | | |
| Thrace | 4 | 0.8 |
| Macedonia | 75 | 14.9 |
| Thessaly | 34 | 6.8 |
| Epirus | 22 | 4.4 |
| Central Greece | 42 | 8.4 |
| Peloponnese | 50 | 10.0 |
| Crete | 70 | 13.9 |
| Ionian Islands | 102 | 20.3 |
| Aegean Islands | 103 | 20.5 |
| **Stars** | | |
| 5 | 18 | 3.6 |
| 4 | 101 | 20.1 |
| 3 | 177 | 35.3 |
| 2 | 133 | 26.5 |
| 1 | 73 | 14.5 |
| **Number of employees** | | |
| 1–10 | 245 | 48.8 |
| 11–50 | 171 | 34.1 |
| 51–250 | 73 | 14.5 |
| >250 | 13 | 2.6 |
| **Rooms quantity** | | |
| Less than 200 rooms | 452 | 90.0 |
| 200–399 rooms | 35 | 7.0 |
| 400–699 rooms | 10 | 2.0 |
| More than 700 rooms | 5 | 1.0 |
| **Annual Income** | | |
| <2 million euros | 245 | 48.8 |
| 2–10 million euros | 172 | 34.3 |
| 11–50 million euros | 77 | 15.3 |
| More than 50 million euros | 8 | 1.6 |
| **Operation mode** | | |
| Full-year | 200 | 39.8 |
| Seasonal | 302 | 60.2 |

### 4.2. Method of Analysis

The current study was conducted through "Partial Least Squares Structural Equation Modeling (PLS-SEM) via Smart-PLS 3.2". In recent years, PLS-SEM has been attracting researchers' interest [64]. For the purpose of this study, PLS-SEM was selected for its ability to assess unobservable variables in the measurement model at the observation level [65]; for testing the hypothesis between latent variables in the structural model in the theoretical level [66]; for testing complex models that include dependent and independent variables, including several in the same model [67]; for its ability to easily incorporate reflective and formative variables [68]; for predictive purposes [64]; and for its high levels of statistical power [69,70]. Concerning the measures, 'Digital Maturity' was measured by 22 items of Gill et al. [44]. 'Financial Resources Availability' was measured by a three-item scale from Wiklund and Shepherd [71] and Story, Boso, and Cadogan [72]. Furthermore, 'Government

Regulation' was measured by three questions that investigated government regulations applied during the pandemic to eliminate the spread of the virus, and 'digital technologies' were measured by eight questions that explored the digital technologies adopted by the hotels during the COVID-19 pandemic.

## 5. Results

### 5.1. Measurement Model

Most commonly, second-order variables are measured using reflective indicators [73,74]. In the current model, first-order reflective factors are used to formulate the second-order variable, which is digital maturity. For the evaluation of the measurement model, it is necessary to assess the internal consistency, convergent validity and discriminant validity via Cronbach's alpha coefficient (a), average variance extracted (AVE) and Fornell and Larcker criterion [75]. The reliability coefficient Cronbach's alpha for all the constructs was within an acceptable level of reliability [76]. The CR and the AVE for all constructs were higher than 0.7 and 0.5, respectively [77] (Table 2). The Fornell–Larcker criterion was used to assess the discriminant validity of the measurement model. More specifically, the correlation coefficients among constructs were lower than the square root of each AVE value [75] (Table 3). In addition, the confidence interval (CI) of Heterotrait–Monotrait ratio of correlations (HTMT) exceeded 1.00, which means that the assumption of discriminant validity was achieved [77] (Appendix B). There is no evidence of collinearity among the items of the digital technology construct as the variance inflation factor (VIF) value was lower than 5. Moreover, the items were considered valid while the outer loadings were more than 0.5 (Table 4) [78].

**Table 2.** Results of the reflective construct assessments.

| Constructs | Items | Factor Loadings | Mean (SD) | CR | AVE |
|---|---|---|---|---|---|
| | DMC1 | 0.835 | 3.33 (0.96) | 0.940 | 0.692 |
| | DMC2 | 0.839 | 3.58 (1.02) | | |
| | DMC3 | 0.827 | 3.12 (1.14) | | |
| Culture (a = 0.924) | DMC4 | 0.877 | 3.13 (1.12) | | |
| | DMC5 | 0.883 | 3.18 (1.16) | | |
| | DMC6 | 0.882 | 3.31 (1.04) | | |
| | DMC7 | 0.659 | 3.9 (0.99) | | |
| Environment | EN1 | 0.669 | 3.22 (1.15) | 0.765 | 0.540 |
| Government regulations | EN2 | 0.685 | 4.27 (0.92) | | |
| (a = 0.774) | EN3 | 0.965 | 3.92 (1.02) | | |
| | DMIN1 | 0.887 | 2.96 (1.15) | 0.953 | 0.801 |
| | DMIN2 | 0.907 | 2.81 (1.15) | | |
| Insights (a = 0.938) | DMIN3 | 0.891 | 3.02 (1.1) | | |
| | DMIN4 | 0.869 | 3.49 (1.12) | | |
| | DMIN5 | 0.92 | 3.35 (1.09) | | |
| | DMOR1 | 0.855 | 3.19 (1.11) | 0.911 | 0.720 |
| Organization (a = 0.869) | DMOR2 | 0.855 | 3.17 (1.05) | | |
| | DMOR3 | 0.909 | 3.37 (1.09) | | |
| | DMOR4 | 0.769 | 3.57 (1.07) | | |
| Organization | FIN1 | 0.903 | 2.90 (1.24) | 0.935 | 0.828 |
| Financial resources | FIN2 | 0.902 | 2.80 (1.16) | | |
| availability (a = 0.896) | FIN3 | 0.925 | 2.78 (1.24) | | |
| | DMTECH1 | 0.811 | 2.91 (1.22) | 0.946 | 0.747 |
| | DMTECH2 | 0.874 | 3.38 (1.09) | | |
| Technology (a = 0.932) | DMTECH3 | 0.902 | 3.17 (1.14) | | |
| | DMTECH4 | 0.915 | 3.18 (1.15) | | |
| | DMTECH5 | 0.796 | 3.78 (1.1) | | |
| | DMTECH6 | 0.881 | 3.19 (1.13) | | |

**Table 2.** *Cont.*

| Constructs | Items | Factor Loadings | Mean (SD) | CR | AVE |
|---|---|---|---|---|---|
| | DMC1 | 0.737 | 3.33 (0.96) | 0.975 | 0.644 |
| | DMC2 | 0.758 | 3.58 (1.02) | | |
| | DMC3 | 0.779 | 3.12 (1.14) | | |
| | DMC4 | 0.814 | 3.13 (1.12) | | |
| | DMC5 | 0.821 | 3.18 (1.16) | | |
| | DMC6 | 0.851 | 3.31 (1.04) | | |
| | DMC7 | 0.654 | 3.9 (0.99) | | |
| | DMOR1 | 0.844 | 3.19 (1.11) | | |
| | DMOR2 | 0.755 | 3.17 (1.05) | | |
| | DMOR3 | 0.837 | 3.37 (1.09) | | |
| Digital maturity (a = 0.973) | DMOR4 | 0.678 | 3.57 (1.07) | | |
| | DMTECH1 | 0.761 | 2.91 (1.22) | | |
| | DMTECH2 | 0.813 | 3.38 (1.09) | | |
| | DMTECH3 | 0.853 | 3.17 (1.14) | | |
| | DMTECH4 | 0.876 | 3.18 (1.15) | | |
| | DMTECH5 | 0.757 | 3.78 (1.1) | | |
| | DMTECH6 | 0.869 | 3.19 (1.13) | | |
| | DMIN1 | 0.851 | 2.96 (1.15) | | |
| | DMIN2 | 0.818 | 2.81 (1.15) | | |
| | DMIN3 | 0.816 | 3.02 (1.1) | | |
| | DMIN4 | 0.804 | 3.49 (1.12) | | |
| | DMIN5 | 0.869 | 3.35 (1.09) | | |

**Table 3.** Discriminant validity (Fornell–Larcker criterion).

| | 1. | 2. | 3. | 4. | 5. | 6. |
|---|---|---|---|---|---|---|
| 1. Culture | 0.832 | | | | | |
| 2. Government Regulations | 0.209 | 0.735 | | | | |
| 3. Insights | 0.795 | 0.178 | 0.895 | | | |
| 4. Financial Resources Availability | 0.270 | −0.065 | 0.243 | 0.910 | | |
| 5. Organization | 0.831 | 0.179 | 0.818 | 0.271 | 0.848 | |
| 6. Technology | 0.829 | 0.192 | 0.870 | 0.279 | 0.851 | 0.864 |

**Table 4.** Results of the formative construct of digital technologies.

| Items | VIF | Outer Loadings | 95% CI | Outer Weights | 95% CI |
|---|---|---|---|---|---|
| Website | 1.674 | 0.586 | 0.484, 0.687 | 0.135 | 0.017, 0.27 |
| Social Media | 1.886 | 0.676 | 0.571, 0.75 | 0.215 | 0.061, 0.339 |
| Mobile and Tablet Applications | 2.033 | 0.797 | 0.711, 0.851 | 0.262 | 0.103, 0.407 |
| QR | 1.738 | 0.709 | 0.603, 0.793 | 0.177 | 0.045, 0.337 |
| Remote Control Systems | 1.545 | 0.626 | 0.503, 0.714 | 0.119 | −0.039, 0.242 |
| Advanced Policy Management | 1.873 | 0.669 | 0.582, 0.738 | 0.08 | −0.036, 0.213 |
| CRM | 1.907 | 0.747 | 0.642, 0.818 | 0.27 | 0.112, 0.432 |
| AI | 1.344 | 0.566 | 0.479, 0.640 | 0.197 | 0.097, 0.29 |

The path-weights of the first-order construct were examined in order to evaluate the significant value on the second-order construct. Furthermore, to check multicollinearity, the VIF was assessed and revealed that all constructs had values lower than 5 and were statistically significant, except for the 'technology' construct where the corresponding value was marginal higher than 5 [79] (Table 5). This outcome is slightly higher but falls in line with the acceptable level of VIF of less than 10 [77].

**Table 5.** Multicollinearity diagnostics and path weights of first-order constructs on the second-order construct.

| Construct | Path Weights | Variance Inflation Factors (VIF) |
|---|---|---|
| Culture | 0.322 [1] | 4.019 |
| Insights | 0.259 [1] | 4.627 |
| Organization | 0.184 [1] | 4.657 |
| Technology | 0.304 [1] | 5.858 |

[1] $p < 0.001$.

### 5.2. Structural Model

The results in Figure 2 present the hypothesized conceptual framework structural model. In addition, Table 6 depicts the outcomes of the model. 'Culture' (b = 0.322, $p < 0.001$), 'insights' (b = 0.259, $p < 0.001$), 'organization' (b = 0.184, $p < 0.001$) and 'technology' (b = 0.304, $p < 0.001$) were significantly correlated with the 'digital maturity', indicating that higher values of independent variables were associated with higher values of 'digital maturity'. Furthermore, 'government regulations' had a positive effect on the construct of digital technologies (b = 0.151, $p < 0.001$). In addition, 'financial resources availability' (b = 0.142, $p = 0.001$) and 'digital maturity' (b = 0.563, $p < 0.001$) had a positive effect on digital technologies.

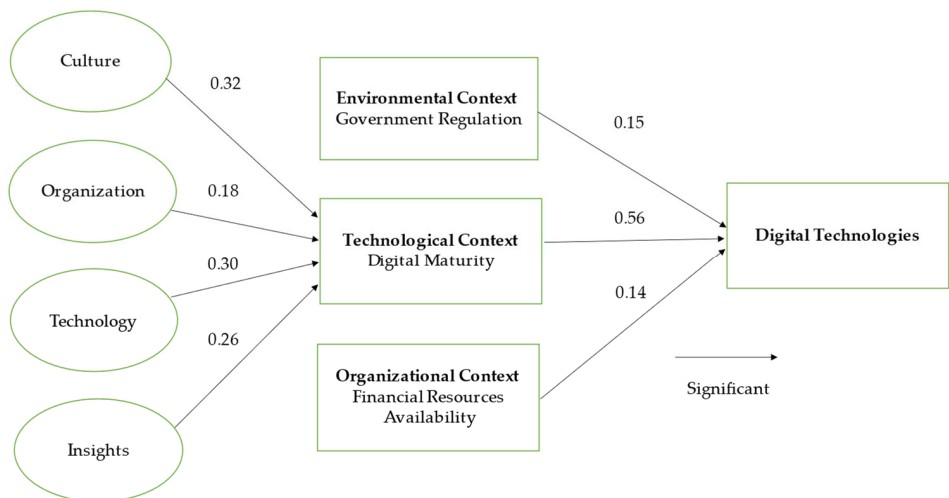

**Figure 2.** Results for the structural model analysis.

**Table 6.** Structural model path coefficients and significance level.

| Hypotheses | Effects | Path Coefficients | t | p | Support |
|---|---|---|---|---|---|
| H1 | Digital Maturity -> Digital Technologies | 0.563 | 16.85 | <0.001 | Yes |
| H1a | Culture -> Digital Maturity | 0.322 | 61.94 | <0.001 | Yes |
| H1b | Organization -> Digital Maturity | 0.184 | 45.34 | <0.001 | Yes |
| H1c | Technology -> Digital Maturity | 0.304 | 66.08 | <0.001 | Yes |
| H1d | Insights -> Digital Maturity | 0.259 | 53.91 | <0.001 | Yes |
| H2 | Financial resources availability -> Digital technologies | 0.142 | 3.46 | 0.001 | Yes |
| H3 | Government regulations -> Digital Technologies | 0.151 | 4.25 | <0.001 | Yes |

In order to assess the predictive power of the model, the coefficient of determination ($R^2$) was calculated. $R^2$ was 0.44 and explained about 44% of the variability in 'digital technologies' [78]. The structural model explains 100% of the variability of 'digital maturity' in digital technologies. The Stone–Geisser's $Q^2$ value was calculated in order to assess the predictive relevance of the model. The $Q^2$ values of 'digital maturity' ($Q^2 = 0.640$) and

'digital technologies' ($Q^2 = 0.196$) were greater than zero, indicating the predictive relevance of the model [80]. The standardized root-mean-square residual (SRMR) value was 0.066, larger than 0.08, which supports good model fit [75].

## 6. Discussion

### 6.1. Digital Maturity as Predictor of Technology Adoption

Digital maturity was found to be a critical factor of digital technology adoption in the hospitality industry, thereby confirming Hypothesis 1 (H1a, H1b H1c, and H1d). Generally, the technological component includes the internal and external technology equipment or process characteristics of a company [12]. This finding implies that hotel owners and managers accentuate their digital strategy implementation when they have achieved higher degrees of digital maturity. In this study, the digital maturity of a firm is decomposed into four dimensions, namely culture, technology, organization and insights [36,44]. Our findings are aligned with past studies, which highlighted that technological readiness, a closely related term with digital maturity, consists of an important determinant of technology adoption [61,62]. More analytically, the 'culture' dimension is defined as the digital-driven approach of the company towards digital advances [44]. Previous examination of the literature has emphasized the technological knowledge [81]. However, the empowerment of the employees and the supportive climate are fundamental in the digital technology adoption process [36]. The empirical outcomes of the current study revealed the great impact of this dimension on digital technology adoption in the hotel sector.

Further in the discussion, the 'organization' dimension, which has been identified as the degree of alignment on digital strategy, governance and execution, is critical for the technology adoption and implementation [36]. In effect, the existence and availability of digital technology infrastructure and trained human capital within a firm may alleviate risks pertaining to the adoption of advanced information and communication technologies [82]. The organization dimension was found to be a factor influencing the digital technology adoption, which supports the hypothesis. This result is in consistent with previous studies [83,84].

The 'technology' dimension had previously been mostly highlighted from the business and technology literature as the technological infrastructure and on the importance of perceived benefits [31,85,86]. However, the budget to finance the digital technology initiatives remains an important factor to successfully adopt them [51]. In the same vein, the customers' perception is pivotal in the digital adoption process [44].Thus, the findings of the study contributed to the ongoing technology and hospitality literature as a countable factor.

Lastly, regarding the 'insights' element, the literature revealed that managers are identified as a prime antecedent of technology adoption in the hospitality industry [21]. Likewise, managerial support may affect the awareness of technology adoption [87]. These outcomes are aligned with the findings of the current study. However, the 'insights' dimension clarifies that successful managerial support has to ensure that the employees understand that their performance is tied to the company's digital goals. Hence, this research has contributed to the hospitality literature. Lastly, there is a positive effect of the customer feedback values for measuring the success of the technology adoption. Overall, digital maturity involves the sophistication level of technology usage in the firm [88].

In our study, we posit that hotels that can identify the value provided from their digital maturity may have greater possibility for adopting digital technologies. Interestingly, the technological dimension in the TOE framework was found to be the most statistically significant predictor of technology adoption. Out of the four technological features, 'culture' is found to be the most significant factor affecting digital technology adoption, while 'technology' and 'insight' follow in significance. 'Organization' is found to be the least significant one compared to those mentioned above.

*6.2. Organizational Elements and Technology Adoption*

Concerning the organizational context, the model demonstrates that the availability of financial resources was significantly and positively associated with the adoption and implementation of digital technologies. Thus, the hypothesis is confirmed. This suggests that hotels that are able to finance their business initiatives in a short-term period or are able to attract financial support when needed, are more likely to adopt digital strategies. Financial resources constitute an important feature acknowledged in the business literature [57]. Implementing digital strategies requires financial investment and commitment [89,90]. In effect, scholars suggest that firms with greater financial commitment are more likely to adopt e-business implementation [89]. Thus, our findings are in line with the outcome of the proposed research model. However, previous results indicate that firms' size is a good predictor of technology adoption in the organizations [81]. Larger firms have more resources and could afford greater risks associated with technology adoption [88]. Further, MSMEs are commonly characterized as organizations with financial constraints that inhibit the digital technology adoption [91]. Consequently, with these barriers, the MSMEs are less likely to absorb the shock of a risky or unsuccessful digital technologies investment [92]. In this study, the vast majority of the hotels are MSME independent hotels. Independent hotels have also limited resources compared to high-end hotels or hotels that are affiliated with hotel chains [93]. The assessment of the intention to adopt digital technology in the hotel business generates additional knowledge on the issue [94].

*6.3. Environmental Elements and Technology Adoption*

In the context of environmental factors, government regulations were found to moderately influence digital technology adoption in the hospitality industry. Hence, the last hypothesis of the study is confirmed. According to the extant literature, government regulations may become either facilitators or inhibitors in the digital technology adoption process [82]. In this study, the existence of favorable government regulations for hotels was perceived to facilitate the technology adoption process, although many studies recognize that competitive pressure is the game-changing factor concerning the adoption of digital technology rather than the governmental regulation itself [14,81]. The finding that government regulation is an inhibitor is aligned with Delmas [95], who suggested that the adoption of the ISO standards has led the firms to experience higher costs but be more likely to adopt technology. In the same line as this study, although the government regulations forced hoteliers to finance the training of their employees to COVID-19 heath protocols, they are still more likely to adopt digital technology. Further, Xu et al. [96] suggested that the environmental factor amongst others government regulations has emerged as one of the fundamental factors shaping e-business adoption. They came to the conclusion that governments could accelerate e-business adoption by establishing supportive regulatory environment in the early stages of e-business development. This outcome is aligned with the current study that supported that the government regulations positively and significantly affected the adoption of digital technologies in the Greek hotel sector.

**7. Research and Practical Implications**

The study adopts the perspective of hotel managers towards digital technology adoption. Previous examination of the hospitality industry literature has revealed a general interest in researching digital technology adoption from the customers' perspective [97], and managers' standpoint on the adoption of digital technologies remains relatively unexplored [14]. Moreover, a sightly smaller number of studies have focused on managers' perspective, although they are the decision makers when it comes to digital technology within the hotel. As such, this study contributes to both the information systems and hospitality literature.

For hotel owners and managers, our study findings offer the opportunity to benchmark their cultural, organizational, technological and insight capabilities and create a starting point for future technology planning. To our knowledge, this study is among the first

ones to empirically explore the effect of organizational digital maturity on technology adoption. Digital maturity is a complex phenomenon, and firms have to address cultural, organizational, technical and insight challenges to achieve their digital transformation [36]. Prior examination of the hospitality literature has investigated digital maturity indirectly through the technology readiness of consumers and their perceptions towards adopting technological innovations [47]. This study adopts the perspective of hotel executives and contributes to the growing body of digital maturity measurement models using the hospitality industry as a context.

In the context of constantly improving business performance together with the increasing demand for more sophisticated and high-quality services in the hospitality industry, it is crucial for hotels to become more reliant on digital technology in their business operations [98]. In effect, studies suggest that the most powerful tool to uplift tourism is the adoption of digital technology [99] and, as such, digital technology innovations, products, and processes could be employed as a transformation tool for the provision of hospitality services during and after the pandemic [100]. Thus, the proposed model could provide valuable evidence to public and private stakeholders in order to make more informed decisions.

Furthermore, this study adopts a multi-investigation perspective pertaining to the adoption of digital technologies; past studies emphasized the investigation of one particular type of technology, such as service robots [101] or mobile applications [2]. Our study explores a range of digital technologies that include the adoption of internet technologies (e.g., social media strategies), inter-organization information systems and emerging technologies, such as artificial intelligence. (For a full list of the examined digital technologies please refer to Appendix C). Hotels have been adopting technologies to optimize services at scale in order to respond to customers' and markets' growing demands [45]. To this extent, different types of technology serve accordingly different purposes within the hotel operation. All in all, the findings provide practitioners with a knowledge of various digital technologies as a whole rather than partially.

Researchers agree that digital technologies will continue to advance in the hospitality industry [98,102]. Meanwhile, crises such as the COVID-19 pandemic have created challenges concerning technology adoption due to social distancing restrictions [103]. Thus, it might be beneficial for hoteliers and managers to stay informed and use the evidence of the current theoretical model to support their business operations and prepare in advance for possible future crises.

## 8. Limitations and Future Research

The study has several limitations that also provide opportunities for future research. The first limitation is that the research examines managers' perceptions towards the adoption of digital technologies, rather than their actual usage of technology. Future research may include field studies that capture actual investments of hospitality industry firms on digital technologies. Second, our empirical sample is restricted to hotels in Greece. Future studies could explore wider populations and different cultures as well. In addition, future research could consider other variables as an extension of the conceptual framework. Since the COVID-19 pandemic is a changing dynamic condition [104], it would be helpful to evaluate findings in the different stages of the pandemic. Longitudinal studies could also be designed to investigate the role of digital technology adoption compared with hotels in the pre- and post-pandemic era to expand the findings and apply various perspectives. Furthermore, different impacts and scenarios could be explored by comparing the outcomes. Thirdly, to gain a more holistic view of the adoption and implementation of digital technologies in the hospitality sector, the investigation of the ambient intelligence era (2020–future), which remains unexplored, could offer a comparison with interesting insights.

Moreover, as only hotels are included in the research and no other hospitality accommodation category, this is considered as a limitation. Although most of the hotels were not in full operation or closed, we managed to contact them. However, the strict

government regulations to secure physical isolation and the suspension of the hospitality industry to limit COVID-19 spread have negatively affected the industry and reduced the response [105,106].

This research study focused on the positives of digital technology adoption. However, a dark side of advanced technology adoption is in existence [98,107]. Specifically, scholars have recently revealed that the COVID-19 pandemic has worsened already challenging and vulnerable situations in the tourism and hospitality sector [106]. Further in this context, future research could be conducted to investigate COVID-19 negative impacts (e.g., on hotel employees' mental and psychological health) as a consequence of the working conditions such as remote working, virtual teams [106] and service automation [27]. Likewise, future research can address issues including indifference for others and problems with worker unions in the hospitality industry.

## 9. Conclusions

Consequently, the objectives of the study are fourfold: Firstly, to identify relevant factors that influence the digital technology adoption in the hotel industry; secondly, to derive a theoretical framework that incorporates these influential measures; thirdly, to propose relevant propositions; and lastly, to discuss the research and practical implications and directions for future research in this field. Digital technology adoption in the hotel sector is an ongoing effort rather than a fixed state as the pace of the market is constantly changing together with COVID-19 turbulence. Therefore, exploring the determinants of technology adoption in the hotel industry is essential to gain a deeper understanding of the process. The findings of the study provide useful insights to hotel owners, managers and policy makers, and highlight approaches for future studies.

**Author Contributions:** Writing—original draft preparation, M.N. and P.K.; supervision: P.K.; data analysis, M.N., P.K., G.C., A.P. and N.M.; methodology; M.N., P.K. and A.P.; review and editing: A.P. and N.M. All authors have read and agreed to the published version of the manuscript.

**Funding:** This research received no external funding.

**Institutional Review Board Statement:** Not applicable.

**Informed Consent Statement:** Not applicable.

**Data Availability Statement:** Not applicable.

**Acknowledgments:** The authors are thankful to the Greek Ministry of Tourism as it approved our request to distribute the e-questionnaire among the hotels, and we are grateful to the hotels that responded.

**Conflicts of Interest:** The authors declare no conflict of interest.

## Appendix A

**Table A1.** Items of research constructs.

| Constructs | | Items |
|---|---|---|
| DIGITAL MATURITY Culture | DMC1 | The business strategy is based on digital technologies |
| | DMC2 | Business management supports the digital strategy of the business |
| | DMC3 | We have the proper staff to manage digital technologies in the business |
| | DMC4 | We invest in the training of our staff for the use of digital technologies |
| | DMC5 | The management of the company communicates its digital strategy to the staff of the company |
| | DMC6 | The company takes all necessary actions to support innovation |
| | DMC7 | We are interested in enhancing the experience of our customers by utilizing different service channels (e.g., through a website, call center, social media, etc.) |

**Table A1.** *Cont.*

| Constructs | | Items |
|---|---|---|
| DIGITAL MATURITY | DMOR1 | We commit the necessary resources (human and/or financial) to design, redefine and execute our digital strategy |
| Organization | DMOR2 | Our staff has the necessary digital skills to use information and communication systems |
| | DMOR3 | Our company follows specific procedures for the management of our information systems |
| | DMOR4 | We use digital channels to communicate with our business partners (e.g., suppliers, banks, etc.) |
| DIGITAL MATURITY | DMTECH1 | We have set a specific budget for the supply/upgrade of our digital infrastructure and systems |
| Technology | DMTECH2 | Our business is flexible in changes related to its digital strategy |
| | DMTECH3 | Our company uses and utilizes modern information systems and infrastructures |
| | DMTECH4 | Our company evaluates the performance of its information systems and infrastructures in terms of their contribution to the achievement of business goals |
| | DMTECH5 | Our company utilizes feedback from its customers to reshape its digital strategy |
| | DMTECH6 | Our company uses digital technologies to promote innovation and collaboration with its staff |
| DIGITAL MATURITY | DMIN1 | Our company has set clear and quantitative targets for measuring the success of its digital strategy |
| Insights | DMIN2 | Every employee of the company understands how their performance is related to specific corporate digital goals |
| | DMIN3 | Our business staff understands the ways in which physical and digital channels work together to achieve the desired result |
| | DMIN4 | The views of the company's customers are used to develop new digital services |
| | DMIN5 | We use the experience from the implementation of our digital actions in the shaping of our digital strategy |
| ORGANIZATION Financial resources availability | FIN1 | If we need financial help for our business activities, we can receive them |
| | FIN2 | We have financial resources to finance our business initiatives |
| | FIN3 | We are able to obtain financial resources in a short period of time to support the operation of our business |
| ENVIRONMENT | EN1 | The training of the staff in the COVID-19 health protocols is a costly undertaking for the hotel |
| Government regulations | EN2 | The compliance with COVID-19 health protocols increases the cost of the hotels' services |
| | EN3 | The alignment with COVID-19 health protocols requires investments to improve hotels' digital infrastructure |

## Appendix B

**Table A2.** CI of HTMT ratio.

| HTMT Ratio | Culture | Government Regulations | Insights | Financial Resources Availability | Organization | Technology |
|---|---|---|---|---|---|---|
| Government regulations | 0.186 (0.11–0.243) | | | | | |
| Insights | 0.852 (0.817–0.882) | 0.162 (0.084–0.215) | | | | |
| Financial resources availability | 0.294 (0.202–0.379) | 0.076 (0.031–0.148) | 0.263 (0.161–0.345) | | | |
| Organization | 0.923 (0.886–0.953) | 0.146 (0.085–0.197) | 0.902 (0.866–0.931) | 0.306 (0.22–0.396) | | |
| Technology | 0.892 (0.859–0.919) | 0.162 (0.086–0.211) | 0.929 (0.899–0.95) | 0.305 (0.217–0.387) | 0.944 (0.919–0.963) | |
| Digital Maturity | 0.984 (0.973–0.994) | 0.173 (0.111–0.217) | 0.971 (0.958–0.981) | 0.304 (0.217–0.382) | 1.002 (0.987–1.016) | 0.998 (0.99–1.007) |

## Appendix C

**Table A3.** Items of digital technologies construct.

| Digital Technologies | Items |
|---|---|
| Website | Enrich website with information on COVID-19 policy (e.g., prevention program, cancellation policy, frequently asked questions about COVID-19) |
| Social Media | Social media campaign/publication series to inform hotel guests concerning COVID-19 policy |
| Mobile & Tablet applications | Development of customer service applications on mobile and tablet (e.g., mobile check-in, communication with the staff via mobile) |
| Quick Response code (QR) | QR codes to avoid the use of printed material (e.g., menu scanning, brochure codes, etc.) |
| Remote Control Systems | Remote control systems (e.g., virtual TV remote control, touchless digital menu) |
| Advanced Policy Management | Advanced policy management (e.g., personalized or dynamic pricing) |
| Customer Relationship Management (CRM) | Advanced customer management systems to improve customer communication and loyalty (CRM) |
| Artificial Intelligence (AI) | Advanced artificial intelligence systems (e.g., robotics systems, guest chatbots, demand forecasting systems) |

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
