# Peer review of "Determinants of Digital Transformation in the Hospitality Industry: Technological, Organizational, and Environmental Drivers"

_sustainability, doi:10.3390/su15032736_

Round 1
Reviewer 1 Report
The topic of the paper is interesting, its purpose is clearly stated, and the problem is well defined. The study is quite good in content and substance, it makes a good the literature review and it uses the correct methodology for this type of study. Furthermore, it makes a good discussion of the results with respect to the studies carried out previously.
Minor directions suggested to the authors to improve the study.
I suggest authors to better introduce the reader to different sections of the paper adding a description of the structure of the paper at the end of the introduction.
In the methodology section, authors should explain about research data such as participants. I suggest giving more details about the criteria for selecting sample and its limits.
In the results section authors should explain better data with reference to the initial hypotheses confirming or not them. I suggest you a more structured presentation of the results as well as a deeper interpretation of the results.
I suggest adding a short paragraph about Conclusion in which you summarise your study and remark your main obtained results.
Author Response
Thank you, please find attached our analytical responses and revisions

Reviewer 2 Report
Hello Authors,
Very worthwhile. Great work. Chapt 2 onwards are of a very high standard. I have particular concerns about Chapt 1 and there are little matters of detail.
Once you have made the changes in the areas I list below I suggest strongly that a person with very strong English language skills assist you to tighten and clarify.
Title:
all good.
Keywords:
add transformation
Introduction:
Does not cover the necessary information.. e.g. the research question, what you mean by digitatal transformation, benefits that the findings reveal, research approach/method.
In intro and elsewhere in Chapt 2, future tense is used. As the research has taken place, the wording is to be changed to past tense.
terms 'digital technology adoption' and 'digital transformation' are used throughout. I think both have a place in your research but in the Intro explain where/how one relates to the other, and that they are different.
Lns 39 and 40. Make clear.
Theoretical background:
Lns 82 to 84.
Ln 115 entirely
Lns 179 to 185 do not provide arygment that justifies H1c.
Ln 196 specifies insights but the text above doe not justify this wording.
Research and Practical Implementations
the quite appropriate discussion of the model created by this research logically leads to the reader looking for a diagram or figure (similar to that in the research menthos section). Creat and label one and that will make your paper more citation worthy.
Over to you. Happy to review your revised paper.
Author Response
Thank you, please find attached our analytical responses and revisions.

Reviewer 3 Report
Technological transformation may benefit too much tourist destinations in these dark days. In a post-COVID-19 world, technology plays a leading role in improving not only client relationships but also in creating smart more sustainable destinations. Anyway, technology has some problems which need to be discussed such as: the depersonalization process, indifference for the Other, problems with worker unions, more costs for employers, and of course the implementation of automatization processes. See furtherly on this literature : Ellul, J. (2021). The technological society. Vintage.- Korstanje, M. E. (2020). Passage from the Tourist Gaze to the Wicked Gaze: A case study on COVID-19 with special reference to Argentina. In International case studies in the management of disasters (pp. 197-211). Emerald Publishing Limited.-Zuccoli, A., & Korstanje, M. E. (2023). Marketing Education in Times of COVID-19: Argentina as Main Study Case. In The Role of Pleasure to Improve Tourism Education (pp. 55-70). Springer, Cham.-Sigala, M. (2020). Tourism and COVID-19: Impacts and implications for advancing and resetting industry and research. Journal of business research, 117, 312-321.-Baggio, R., Micera, R., & Del Chiappa, G. (2020). Smart tourism destinations: a critical reflection. Journal of Hospitality and Tourism Technology.- Li, J., Pearce, P. L., & Low, D. (2018). Media representation of digital-free tourism: A critical discourse analysis. Tourism Management, 69, 317-329.
The strategies to follow in a future agenda is not clear, rewrite please accordingly
Author Response

(The authors gave the same response as above.)

Reviewer 4 Report
At the very beginning, I would like to praise an interesting topic of research, which is increasingly relevant in the world of tourism and catering. We are well aware of the conflicting views on the acceptance of innovative technology in business in the service industry.The abstract is structured according to all standards that are acceptable, as well as the number of key words is quite sufficient and fully answers the description of the manuscript and the problem.
The introductory section is short, but to the point, and provides key introductory information about the manuscript and persistence. That part follows on from the review of literature and similar studies, and accordingly the length of the introductory part is fine. The authors have formulated the hypotheses in the theoretical part and presented the research model very clearly.
The results they obtained were presented very clearly in tabular and graphical form, and at the end they presented the hypothesis scheme and their confirmation graphically at the end of the part that talks about the results. It is a good approach to presenting hypotheses to readers in a concise and simple way.
However, my suggestion would be to move the graphical model of the hypotheses of Figure 1 (where the + sign will not stand), to the part where they set the hypotheses, because in this chapter of the results they provided a table with the results of the path analysis together with the confirmation of the hypotheses, again. Perhaps it would be more clear and not so confusing if the graphic representation of hypotheses in Figure 1 (without confirmation) and panel 1 (with confirmation of hypotheses) were separated.
The discussion and concluding remarks are clear, transparent, extensive and quite sufficiently show the summarized results and the achieved goal of the research. Theoretical and practical implications are given in the concluding part of the manuscript, as well as limiting circumstances.
I suggest literature
Gajić, T., Radovanović, M., Tretiakova, T., Syromiatnikova, J. (2020). Creating brand confidence to gastronomic consumers through social networks – a report fromNovi Sad. Journal of Place Management and Development, Vol. 14 No. 1, pp. 32-42. https://doi.org/10.1108/JPMD-04-2020-003
After the small changes that have been suggested, I propose to revise the English language and the manuscript could be published.
Author Response

(The authors gave the same response as above.)
